# *Poly-FEVER*: A Multilingual Fact Verification Benchmark for Hallucination Detection in Large Language Models

## Abstract

We present *Poly-FEVER*, a large-scale multilingual benchmark for fact verification and hallucination detection in large language models (LLMs). *Poly-FEVER* extends FEVER, Climate-FEVER, and SciFact to 77,973 labeled claims across 11 languages—English, Chinese, Hindi, Arabic, Bengali, Japanese, Korean, Tamil, Thai, Georgian, and Amharic—curated to preserve logical equivalence across scripts and morphologies and to focus on verifiable Supported/Refuted cases. We augment each claim with topic metadata derived from a 22-topic LDA model to enable topic-aware evaluation. Claims are translated using Google Cloud Translation and validated with GEMBA scores averaging 90 across languages, supporting high semantic fidelity. Using *Poly-FEVER*, we benchmark ChatGPT-3.5, LLaMA-2 (7B/13B/70B), and LLaMA-3.1-8B under language-wise, general, and classification prompt families, and study self-detection via rephrasing. We further probe resource imbalance by correlating accuracy with automated web presence (Google hit counts) and test retrieval-augmented generation via DPR over Wikipedia. Results show pronounced cross-lingual disparities: high-resource languages (e.g., English, Chinese) achieve the strongest accuracy, while lower-resource languages (e.g., Amharic, Tamil) lag. Topic structuring consistently benefits lower-resource settings, and RAG provides selective gains (notably in Arabic and Amharic), but can conflict with strong internal priors in high-resource languages. *Poly-FEVER* establishes a rigorous, publicly available foundation for responsible, language-adaptive evaluation of hallucination mitigation in LLMs.

## 1 Introduction

Large language models (LLMs) are now routinely deployed across high-stakes domains—education, healthcare, and law among them—where the factual integrity of generated content is paramount. Yet, mainstream LLMs are trained on corpora that are demographically and linguistically imbalanced (Shah et al., 2019; Li et al., 2023). Language, a salient axis of demographic diversity, remains comparatively under-examined in hallucination detection, raising concerns about fairness and equitable usability across linguistic communities.

Most prior work investigates hallucinations primarily in high-resource languages such as English (Yao et al., 2023), Chinese (Cheng et al., 2023), and German (Sennrich et al., 2023). These studies have driven progress in mitigation via prompt and system design—e.g., retrieval-augmented prompting (Lewis et al., 2020), feedback loops, and prompt tuning—as well as via modeling advances that reduce hallucination propensity through architectural changes, loss design, and supervised fine-tuning (Tonmoy et al., 2024). However, the field lacks a unified, multilingual evaluation that uses the same inputs across languages. As a result, it is difficult to draw systematic conclusions about cross-lingual hallucination behavior or to assess fairness in model performance.

The resulting imbalance is illustrated in Figure 1, which contrasts (A) the global distribution of native speakers (Dunn, 2020; Paolillo & Das, 2006) with (B) the research concentration in a few high-resource languages (Joshi et al., 2020; Yu et al., 2022; Ranathunga & De Silva, 2022), and presents (C) the coverage of our proposed benchmark by resource group and language. This di-

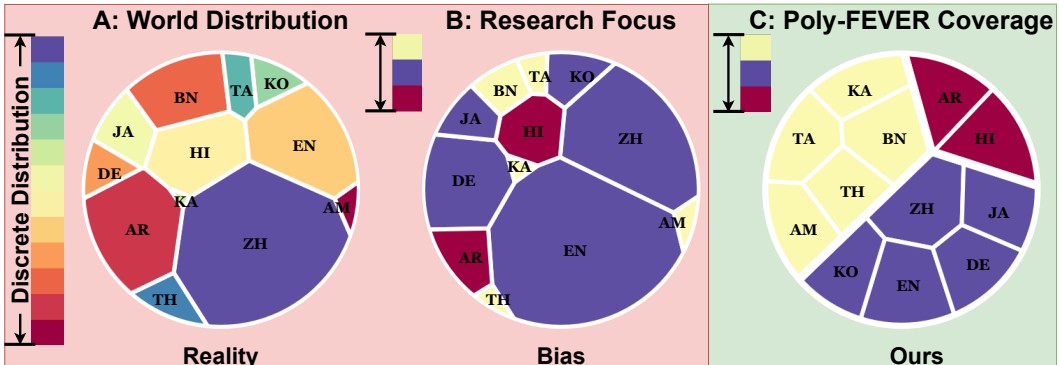

Figure 1: **(A) World distribution:** each polygon corresponds to a language, with area proportional to its share of native speakers, colored uniquely by language. **(B) Research focus:** areas represent how much existing hallucination research concentrates on each language, with colors indicating *High-, Medium-, or Low-resource* categories. **(C)** *Poly-FEVER* **coverage:** hierarchical treemap where outer regions are resource groups and inner polygons are languages, with area proportional to their share in the dataset and colors mapped to resource groups.

vergence underscores the need for a controlled, multilingual testbed to study hallucinations beyond English-centric settings.

A central conceptual distinction in this work is between fact verification and hallucination detection. Fact verification determines whether a claim is *supported or refuted* by established knowledge sources and thus presumes access to external evidence (Murayama, 2021; Zhu et al., 2021). By contrast, hallucination detection targets model-generated falsehoods or inconsistencies, irrespective of whether external references are consulted. Multilingual evaluation further complicates this picture: models operate under heterogeneous data distributions, resource availability, and linguistic structures. Ensuring cross-language consistency therefore requires more than translation; it demands preserving *logical equivalence* across scripts and morphologies so that labels remain faithful to ground truth.

We address these gaps by introducing *Poly-FEVER*, a multilingual benchmark that extends fact verification into a controlled setting for studying hallucination detection across languages. *Poly-FEVER* is designed as a *normalization resource*: it preserves logical consistency of claims across scripts and morphologies, balances representation across languages, and enables structured comparisons of model behavior without relying on ad-hoc, language-specific inputs.

Our contributions are threefold:

1. We release *Poly-FEVER*, an extensive, publicly available dataset for multilingual fact extraction and verification, spanning 77,973 labeled claims across 11 languages, curated for hallucination-detection tasks.
2. We benchmark hallucination detection in contemporary LLMs (ChatGPT-3.5, LLaMA-2 7B/13B/70B, and LLaMA-3.1-8B) under standardized language-wise and classification prompts to quantify cross-lingual disparities.
3. We investigate *why* hallucinations vary across languages by combining topic-aware analysis via Latent Dirichlet Allocation (LDA) with automated web-presence estimates to probe resource imbalance as a driver of performance gaps.

## 2 RELATED WORK

LLMs generate hallucinations—outputs that contradict known facts or fabricate non-existent information, posing challenges for reliable AI applications. Huang et al. (2023) categorize hallucinations into two types: intrinsic and extrinsic. Intrinsic hallucinations occur when a model produces **self-contradictions or logical inconsistencies within its own response**, while extrinsic hallucinations involve **factually incorrect statements that do not align with established knowledge sources**. Evaluating hallucinations requires a systematic comparison between generated content and verified

ground-truth data or an analysis of internal inconsistencies within model outputs. Effective hallucination detection plays a crucial role in ensuring the credibility of generative AI systems, particularly in high-stakes applications such as healthcare, legal services, and scientific research.

Existing evaluation metrics measure hallucination frequency by assessing entity correctness, factual consistency, and contradiction detection. Early methods relied on n-gram overlap (e.g., ROUGE, PARENT-T) (Lin, 2004; Wang et al., 2020), but these metrics fail to capture factual accuracy. Entity-based hallucination metrics (Nan et al., 2021) and relation-based fact extraction models (Goodrich et al., 2019) provide more structured approaches. However, these methods primarily target English-language outputs, limiting their applicability to multilingual hallucination analysis.

To assess hallucination detection, researchers rely on fact-checking datasets such as FEVER (Thorne et al., 2018), Climate-FEVER (Diggelmann et al., 2020), and SciFact (Wadden et al., 2020). These datasets evaluate model responses by comparing claims against external sources. However, they primarily focus on fact verification, not hallucination detection, since they assume that relevant evidence is always available.

Current benchmarks also exhibit strong language biases. Most fact-checking datasets contain English-only claims, restricting their ability to evaluate how LLMs handle fact verification across different linguistic contexts. While some efforts attempt multilingual fact-checking (Gupta & Srikumar, 2021), these datasets typically do not provide identical claims across languages, making it difficult to access cross-lingual hallucination patterns.

Most research on multilingual hallucinations focuses on machine translation errors, where LLMs generate hallucinated translations (Pfeiffer et al., 2023). However, translation-based hallucination benchmarks do not evaluate self-contained factual inconsistencies within LLM outputs.

LLMs trained on English-centric corpora often exhibit language-specific hallucination patterns, but existing datasets do not provide a systematic, multilingual benchmark for evaluating hallucination detection across languages. A benchmark that contains the same factual claims across multiple languages, evaluates hallucination detection directly, and exposes cross-linguistic biases is missing from current research.

*Poly-FEVER* fills this gap by introducing a large-scale multilingual benchmark for hallucination detection in LLMs. It extends FEVER, Climate-FEVER, and SciFact to 11 languages, covering 77,973 factual claims.

## 3 *Poly-FEVER* BENCHMARK

This section details the construction, structure, and application of the *Poly-FEVER* benchmark. We describe the dataset curation process, present its organizational structure, analyze the distribution of topics within the data, and evaluate hallucination detection across languages using a range of LLMs.

### 3.1 *Poly-FEVER* OVERVIEW

*Poly-FEVER* provides a multilingual benchmark for fact verification and hallucination detection in large language models (LLMs). It contains 77,973 labeled claims in 11 languages. The dataset extends three widely used English fact-checking sources: FEVER (Fact Extraction and VERification) (Thorne et al., 2018), Climate-FEVER (Diggelmann et al., 2020), and SciFact (Wadden et al., 2020). These sources establish a structured evaluation framework for fact verification by defining claims and classifying them based on supporting or refuting evidence.

FEVER contains 185,445 claims derived from Wikipedia content. The dataset assigns each claim one of three labels: *Supported*, *Refuted*, or *NotEnoughInfo*. The first two categories indicate whether verifiable evidence exists, while the third highlights instances where information remains insufficient. *Poly-FEVER* excludes claims labeled as *NotEnoughInfo* to focus on cases where factual accuracy can be measured directly.

Climate-FEVER builds on the FEVER framework but focuses on climate-related claims. The dataset includes statements verified against scientific literature and expert-reviewed sources. It provides a structured evaluation of fact verification in a domain that requires specialized knowledge. *Poly-*

*FEVER* incorporates only *Supported* and *Refuted* claims from Climate-FEVER to maintain consistency across sources.

SciFact contains scientific claims extracted from biomedical literature. The dataset evaluates whether scientific papers *Support* or *Refute* a given claim. SciFact ensures that the benchmark covers technical content that requires precise verification. By including these claims, *Poly-FEVER* enables a structured evaluation of hallucinations in scientific domains.

The dataset spans diverse topics, including Arts, Music, Science, Biology, and History, ensuring comprehensive fact verification across a wide range of subject areas. These topics vary in specificity and contextual complexity, allowing for an in-depth analysis of how LLMs handle factual claims across different domains. The multilingual nature of *Poly-FEVER* further adds to its complexity, as it introduces variations in linguistic structure, requiring models to process and verify claims across languages with distinct grammar, syntax, and lexical properties. By incorporating multiple languages, *Poly-FEVER* facilitates a more nuanced evaluation of LLMs' ability to distinguish between factual and hallucinatory content across linguistic and cultural boundaries. Through its structured and diverse dataset, *Poly-FEVER* supports research in hallucination detection by providing a benchmark that enables systematic assessment of factual consistency, model biases, and the effectiveness of mitigation strategies across different linguistic contexts.

## 3.2 STRUCTURE OF *Poly-FEVER*

To ensure continuity with prior work and ease of adoption, *Poly-FEVER* mirrors the FEVER family structure while adding multilingual and topic-aware fields. Each entry in *Poly-FEVER* contains four primary fields:

- **ID:** A unique identifier assigned to each claim, allowing for consistent referencing and seamless integration with other annotations or future expansions.
- **Claim:** The textual content of the claim, is provided in 11 distinct languages. This multilingual format highlights the dataset's cross-lingual capabilities, enabling evaluations of how models handle claims across diverse linguistic and cultural contexts.
- **Label:** An annotated veracity label (*true* or *false*) indicating whether the claim aligns with established factual evidence. This ensures that each claim is directly tied to a definitive truth value, reflecting its correctness within the dataset.
- **Topic Distribution:** A vector of the top five predicted topics for the claim, derived through thematic analysis (e.g., LDA). This field offers contextual clues about the claim's domain (e.g., sports, history, science) and aids in cross-topic assessments of model performance.

This four-field structure, especially the multilingual claim component, supports robust assessments of fact-verification methods in varied linguistic settings. It aligns with our core research aim: evaluating language model performance and hallucination tendencies when tasked with fact-checking across different topics, domains, and languages.

## 3.3 LDA ON TOPIC DISTRIBUTION

To analyze hallucination patterns in multilingual fact verification, we apply Latent Dirichlet Allocation (LDA) (Blei et al., 2003), a probabilistic model that represents each claim as a mixture of latent topics and each topic as a distribution over words. Based on coherence scores, we select 22 topics spanning diverse domains, which serve as metadata for evaluating how thematic structure interacts with hallucination frequency. Section 5.5 presents case studies using these topic assignments, while Appendix A.3 provides preprocessing and modeling details.

## 3.4 LANGUAGE SELECTION

We deliberately select 11 languages that span high-resource to low-resource, alphabetic to logographic scripts, and diverse grammatical families, enabling systematic study of linguistic bias. The dataset covers English (en), Mandarin Chinese (zh-CN), Hindi (hi), Arabic (ar), Bengali (bn), Japanese (ja), Korean (ko), Tamil (ta), Thai (th), Georgian (ka), and Amharic (am). These languages appear in order of the number of native speakers. The selection ensures that the benchmark evaluates LLM performance across linguistic groups with different structures, resources, and usage patterns.

Each language presents challenges that influence hallucination patterns in LLMs. Mandarin Chinese, Arabic, and Japanese include logographic or syllabic writing systems that differ from English's alphabetic structure. Tamil, Thai, and Georgian introduce distinct syntactic rules that affect sentence formation. Ambiguity, polysemy, and homophones in Hindi, Korean, and Bengali require models to resolve multiple interpretations of words in context. Amharic and Tamil contain complex morphological patterns that affect tokenization and semantic understanding.

Dialects and sociolects create additional challenges in fact verification. Hindi and Arabic contain regional variations that influence word choice and sentence construction. Korean and Japanese incorporate honorifics and context-dependent expressions that require models to infer meaning from cultural and social cues. LLMs trained primarily on English and widely spoken languages struggle to adapt to these variations, leading to inconsistencies in hallucination detection.

The selection also considers differences in script and orthography. Non-Latin scripts appear in Chinese, Arabic, Thai, Hindi, Bengali, Tamil, Georgian, and Amharic. These scripts require different tokenization strategies, which affect how models encode and process text. The dataset ensures that evaluations reflect performance across writing systems with unique linguistic properties.

### 3.5 Multilingual Claim Translation

We extended the benchmark to 11 languages by translating English claims. Several methods were considered, including Google Cloud Translation, DeepL, and LLM-based translators.

Several coauthors fluent in multiple languages assessed translation quality by reviewing selected claims in two languages. Our manual audit covered 1% of the dataset, with two annotators reviewing each sampled claim independently (i.e., each claim was inspected twice). The evaluation focused on preserving cultural and contextual meaning. Google Translate produced more accurate translations than other methods. DeepL Translate introduced structural errors, such as altering verb tense and misordering words. One example included the mistranslation of "The book was read by him" as "Book by him read," which changed the grammatical structure. These inconsistencies led to the exclusion of DeepL and other APIs.

Concerns about **hallucinations in LLM-based translations** influenced the decision to use Google Translate. LLMs generate fluent text but may introduce fabricated details when translating factual claims. Google Cloud Translation provided greater consistency by following source structures without generating new information.

The translation covered over 80,000 English claims into 10 target languages, with a total cost of $2,644 using Google Cloud Translation. To assess quality, we applied the GPT Estimation Metric Based Assessment (GEMBA) (Kocmi & Federmann, 2023) to 5% of the data. GEMBA provides reference-free scores of meaning preservation, and results averaged above 90 across languages (Table 1). These scores indicate that *Poly-FEVER* maintains high semantic fidelity while minimizing translation-induced hallucinations, providing a reliable basis for cross-lingual evaluation.

| Language | Chinese | Hindi | Arabic | Bengali | Japanese | Korean | Tamil | Thai | Georgian | Amharic |
|---|---|---|---|---|---|---|---|---|---|---|
| **AveScore** | 91.3 | 92.4 | 90.8 | 91.8 | 91.5 | **93.0** | 90.0 | 91.1 | 90.8 | 88.9 |

Table 1: Average scores for each language on translation quality evaluation of 5% *Poly-FEVER* benchmark. A score of zero means no meaning preserved.

## 4 Hallucination Causes Exploration

Hallucination behavior in LLMs emerges from an interplay of model internals (training distribution, decoding strategy), external resource availability, and prompt specification. To understand these variations, we examine hallucination detection through three perspectives. First, we evaluate how models classify factual claims under different prompts and whether they recognize their own hallucinations. Second, we analyze external factors, such as the availability of references on the web, which may contribute to hallucination rates. Third, we assess whether retrieval-augmented generation (RAG) improves factual accuracy by providing external evidence. Together, these analyses reveal key factors affecting hallucination patterns in multilingual fact verification.

Figure 2: A conceptual overview illustrating the components of the LLM inference process that may influence hallucination behavior, highlights how multilingual claims interact with tokenization, embeddings, retrieval, prediction, and self-detection, as well as external factors such as web presence and prompt type. Multilingual claims and prompts ($x$) are preprocessed through normalization and tokenization, then passed into the LLM network. **Web presence** provides external reference bias by measuring search-hit counts as a proxy for training exposure. **Retrieval** encodes claims into dense embeddings, retrieves top-$k$ Wikipedia passages, and concatenates evidence with the claim. The LLM generates candidate outputs ($y_t$) using token-level probabilities ($P(y_t \mid y_{<t}, x)$) and greedy decoding. A **self-detection loop** rephrases claims to assess prediction consistency.

## 4.1 MODEL PERFORMANCE

### 4.1.1 EFFECT OF PROMPT DESIGN

LLMs often drift from strict binary labels; prompt design is therefore crucial. We use three prompt families—*General*, *Language-wise*, and *Classification*. The **General** prompt gives a minimal binary instruction; the **Language-wise** prompt (our default) enforces `true`/`false` in the claim's language; the **Classification** prompt maps free-form outputs to a binary label. We show only the Language-wise prompt here because it yielded the most reliable compliance across languages. Full verbatim prompts are in Appendix B.1.

---

**Language-wise Prompt for Multilingual Evaluation**

Assess the truthfulness of the user's claim and provide a response. Use *true* to indicate that the claim is true, *false* to indicate that it is false. Your response should only consist of *true* or *false*, without any additional characters or punctuation.

---

### 4.1.2 LLM-REPHRASED CLAIMS AS A PROXY FOR INTRINSIC HALLUCINATIONS

We address a concern regarding the capacity of LLMs to identify and mitigate hallucinations in the text they generate. Despite the construction of a *Poly-FEVER* dataset established at detecting hallucinations, a gap exists in the literature concerning the effectiveness of these models in recognizing inaccuracies within their own outputs. This gap stems from the fact that the claims and labels in *Poly-FEVER* are not produced by LLMs, raising questions about the representativeness of the hallucinations that LLMs themselves produce.

---

**Rephrase Prompt for Synthetic Claim Generation**

Rephrase the following claim without changing its meaning. Ensure the essence and intent remain unchanged.

---

To bridge this gap, we instruct LLMs to rephrase dataset claims in multiple languages while keeping their original meaning intact. Claims generated by LLMs lack verifiable ground truth, which is essential for systematically assessing the model's hallucination detection accuracy. To approximate intrinsic hallucinations under controlled conditions, we rephrased all 77,973 claims through LLM generation while anchoring to their original veracity labels. This yields synthetic outputs that mimic model-produced text yet remain evaluable against ground truth. Due to the lack of automated metrics, a manual audit of 100 English and Chinese claims confirmed semantic fidelity, validating this proxy for intrinsic hallucinations. By doing this, we establish to evaluate whether LLMs can detect hallucinations in their own generated text with similar accuracy to their performance.

## 4.2 WEB SEARCH ON REFERENCES BIAS

As LLMs are black boxes for users, we utilized a Python-based automated web scraping tool to examine the presence of claims in *Poly-FEVER* on the web across 11 selected languages. This examination establishes to identify potential biases in training datasets that cause imbalanced performance on the multilingual fact-checking task.

To simulate varied internet user environments and bypass potential search engine restrictions, we randomized user-agent strings and inter-query intervals to avoid rate-limiting and anti-bot filtering Olteanu et al. (2019); Kulshrestha et al. (2017) so that all languages are sampled comparably, emulating heterogeneous internet access patterns to reduce sampling bias and evade anti-bot detection. Further enhancing the authenticity of our approach, we introduced randomized time intervals between search queries, mimicking human browsing behavior and avoiding anti-bot mechanisms.

We performed web searches to operationalize web presence as Google search hit counts, using them as a proxy for training exposure and resource imbalance across languages. This approach allows for a nuanced understanding of how widely each claim is disseminated across different linguistic contexts on the web. The detailed analysis is in section 5.4.

## 4.3 EVALUATING EXTERNAL KNOWLEDGE FOR HALLUCINATION REDUCTION

To test whether external grounding reduces hallucination, we apply Dense Passage Retrieval (DPR) mechanism (Karpukhin et al., 2020). DPR combines a generative model with a dense retriever, allowing claims to be augmented with semantically aligned passages from an external corpus. In our setting, claims are paired with top-$k$ retrieved Wikipedia passages, providing context beyond the model's internal parameters. Section 5.5 and Case Study E analyze how this augmentation interacts with topic awareness. Full retrieval pipeline details are provided in Appendix B.2.

## 5 *Poly-FEVER* CASE STUDIES

### 5.1 SETUP

We report *baseline case studies* under fixed prompts and decoding conditions to illustrate multilingual failure modes and sensitivities. All results are reproducible and representative, obtained without parameter tuning. We evaluate ChatGPT 3.5, LLaMA-2 (7B/13B/70B), and LLaMA-3.1 8B as reference baselines using exact-match accuracy over binary truth labels with language wise prompts and low-temperature decoding unless otherwise stated. Full hardware/software stacks are provided in Appendix C.1.

### 5.2 CASE STUDY A: TEMPERATURE SENSITIVITY

We include an illustrative sweep for LLaMA-3.1 8B across temperatures $\{0.1, \ldots, 2.0\}$ to show language-specific response patterns under fixed prompts (Appendix C.2, Table 7). English trends upward with higher temperatures, while Chinese and Amharic decline; several languages (e.g., Japanese, Bengali) are comparatively stable.

### 5.3 CASE STUDY B: MULTILINGUAL BASELINES FOR FACT VERIFICATION

| Lang. | GPT | GPT Self-Detection | LLaMA-2 7B | LLaMA-2 13B | LLaMA-2 70B | LLaMA-3.1 8B |
|---|---|---|---|---|---|---|
| en | **65.89%** | **61.88%** | **63.35%** | **64.27%** | **64.56%** | 48.43% |
| zh-CN | 58.25% | 53.94% | 58.29% | 59.88% | 38.75% | 54.51% |
| hi | 52.90% | 58.10% | 48.59% | 54.33% | 45.68% | 47.94% |
| ar | 45.48% | 58.80% | 50.55% | 54.97% | 32.97% | 49.33% |
| bn | 55.65% | 58.73% | 49.05% | 52.30% | 33.87% | 45.86% |
| ja | 55.89% | 58.95% | 58.24% | 59.57% | 41.37% | **55.34%** |
| ko | 57.29% | 60.14% | 56.67% | 58.74% | 46.06% | 52.47% |
| ta | 55.67% | 59.98% | 49.54% | 50.86% | 19.47% | 53.18% |
| th | 56.82% | 52.04% | 53.90% | 53.46% | 48.09% | 50.37% |
| ka | 57.37% | 51.69% | 47.39% | 53.45% | 46.15% | 48.71% |
| am | 47.53% | 47.06% | 43.42% | 48.64% | 26.34% | 48.37% |

Table 2: Comparison on accuracy of hallucination detection on fact-checking task by ChatGPT 3.5, ChatGPT 3.5 Self-Detection, LLaMA-2 7B, 13B, 70B and LLaMA-3.1 8B. Highest values bolded, lowest values underlined.

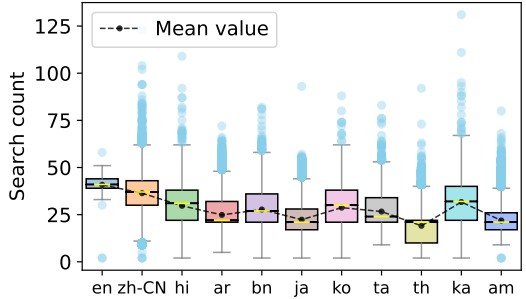

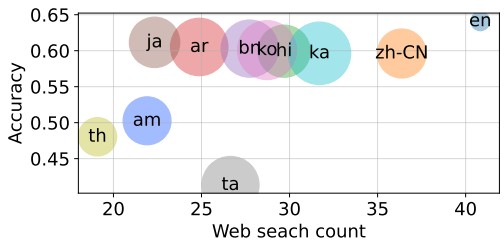

Figure 3: Web search distribution on multilingual claims. Middle 50% of search counts inside each box, mean values are connected.

Figure 4: Detection accuracy on web search count. The x-axis reflects average search-hit counts, and the y-axis shows hallucination detection accuracy. Bubble sizes encode the variance-to-mean ratio of search counts.

We deployed the same fact-checking process on ChatGPT 3.5, LLaMA-2 series to instantiate multilingual baselines on multilingual claims. Specifically, we observed the self-detection ability of ChatGPT 3.5 by prompting it to rephrase the original claims and identify the validity of the generated context. Appendix C.3 reports results separately for FEVER, Climate-FEVER, and SciFact. In Table 2, English generally achieves higher accuracy than most other languages across models. Self-detection (Rephrase+Classify) shifts vary by language: for some (e.g., Hindi), it improves consistency, whereas English often remains strongest with direct classification. These deltas are diagnostic of prompt-format sensitivity rather than tuned improvements.

### 5.4 CASE STUDY C: WEB PRESENCE AND RESOURCE IMBALANCE

We estimate web presence per language using an automated search procedure (details in Section 3.3). This serves as a coarse proxy for resource availability that may relate to model behavior.

We compare the count of search results across 11 different languages in Figure 3, with an emphasis on identifying potential biases in the training datasets. Some languages, like English, demonstrate a relatively wide interquartile range (IQR), which contains the middle 50% of the data, indicating a high variability in the search count. Others, like Thai, have a much narrower IQR, indicating less variability. Languages like Amharic and Georgian have lower median and mean search counts, indicating less available content or fewer search results for these languages. This disparity could lead to an imbalanced performance in multilingual fact-checking, with better results in languages that have more content available, like English, and worse results in languages with less content.

The relationships between web-search counts and hallucination-detection accuracy across languages are shown in Figure 4. We observe a moderate but non-significant association between the two ($\rho \approx 0.49$, $p = 0.13$). Languages with higher web-search frequencies, such as English and Chinese, tend to show higher accuracy, whereas lower-resource languages, such as Amharic and Tamil, exhibit lower overall accuracy. These patterns should be interpreted descriptively: web presence is one plausible factor that may relate to accuracy differences, consistent with prior work Pires et al. (2019); Singh et al. (2023), but other linguistic and modeling factors may also contribute.

### 5.5 CASE STUDY D: TOPIC SENSITIVITY

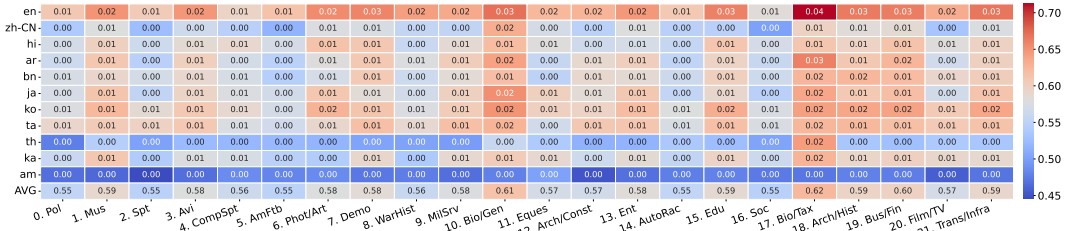

Figure 5: Average accuracy across topics by language, with standard deviation indicated by annotation. The color scale maps accuracy values from low (blue) to high (red), with intermediate values shown in lighter shades. Standard deviation values illustrate the degree of variability within each topic. The bottom row aggregates the mean accuracy and standard deviation for each topic across all languages, providing an overall assessment of topic difficulty and model robustness.

| Lang. | Poli | Sport | Comp | Football | WarHist | Equestr | ArchConst | AutoRace | Soccer | FilmTV |
|---|---|---|---|---|---|---|---|---|---|---|
| zh-CN | 2.62 | 2.5 | 1.17 | 4.72 | 2.78 | 3.02 | 1.02 | 1.28 | 2.66 | 2.59 |
| hi | 1.97 | 0.14 | -1.33 | 2.80 | 4.66 | 1.87 | 0.88 | 0.62 | -0.39 | 1.52 |
| ar | 1.57 | 0.42 | 0 | 4.99 | 0.91 | 1.47 | 0.73 | 1.99 | 0.29 | 1.64 |
| bn | 0.86 | 0.28 | -0.81 | 3.31 | 2.16 | 3.75 | -0.58 | -0.66 | 0.09 | -1.13 |
| ja | -0.35 | -1.81 | -1.38 | 1.11 | 1.21 | -2.85 | 0.22 | 0.43 | 0.35 | 0.21 |
| ko | 0.4 | -2.36 | -0.76 | 1.08 | 1.87 | -0.98 | 0.22 | 0.66 | -0.83 | -0.32 |
| ta | -1.41 | -4.31 | -0.48 | 1.05 | -1.62 | 0.33 | -0.95 | -1.89 | -1.98 | -3.00 |
| th | 6.95 | 6.05 | 2.9 | 4.09 | 6.65 | 4.07 | 6.43 | 2.27 | 3.79 | 3.93 |
| ka | 2.28 | 2.57 | -0.6 | 2.83 | 4.16 | 1.39 | 0 | 0.76 | 0.67 | 0.25 |
| am | 7.71 | 7.44 | 3.76 | 5.05 | 6.98 | 4.24 | 5.41 | 3.60 | 4.41 | 7.09 |

| Lang. | Original | RevPrompt | LDA | LDA+RAG |
|---|---|---|---|---|
| en | **65.89%** | 64.09% | 63.48% | 60.5% |
| zh-CN | 58.25% | 59.61% | 57.81% | 53.74% |
| hi | 52.90% | 59.93% | 60.03% | 54.49% |
| ar | 45.48% | 60.52% | 59.71% | 55.67% |
| bn | 55.65% | 60.29% | 59.57% | 57.35% |
| ja | 55.89% | 61.16% | 58.44% | 55.44% |
| ko | 57.29% | 60.08% | 60.46% | 57.42% |
| ta | 55.67% | 41.34% | 58.83% | 56.89% |
| th | 56.82% | 48.04% | 57.39% | 53.37% |
| ka | 57.37% | 59.64% | 59.44% | 56.99% |
| am | 47.53% | 50.26% | 54.24% | 53.49% |

Table 3: Percentage improvement of correct judgment on hallucination detection. Topic names are shown in English; claims remain in their original languages. Positive values indicate improved accuracy, while negative values indicate a decrease.

Table 4: Accuracy of hallucination detection with original process, prompts revised, LDA, LDA+RAG

We analyze topic-level sensitivity using LDA over 22 topics and 11 languages (Section 3.3).

LLMs generally perform best in English and struggle with Amharic and Thai as shown in Figure 5. Topics such as Politics, Sports, Film/Television, and Warfare History prove challenging across languages due to their subjective nature, where personal biases and interpretations can obscure the distinction between fact and opinionMahl et al. (2024); Hrckova et al. (2022). The dynamic nature of these fields, coupled with the need for specialized knowledge in areas like Architecture/Construction and Competitive Sports, complicates fact-checking. Historical contexts in Warfare History and Automotive Racing add another layer of complexity, as historical records can be biased or incomplete. Emotional ties to topics like American Football and Film/Television can bias information, while the subjective interpretation of data in Sports or Business/Finance makes objective verification difficult. The absence of universal standards in evaluating greatness in sports or the arts further complicates claim verification.

The standard deviation reveals varying degrees of biased hallucinations across languages. While topics 6, 7, 9, 10, 13, 15, and 17 demonstrate high average accuracy, there is significant variance, with languages like English and Chinese exhibiting higher accuracy. This variance underscores how hallucination biases differ among languages, reflecting the complex interplay between linguistic context and the accuracy of LLM predictions on specific topics.

Using LDA for linguistic topic extraction, Table 3 presents the improvements in accuracy for non-English languages such as Chinese, Arabic, Thai, and Amharic across nuanced topics 0, 2, 4, 5, 8, 11, 12, 14, 16, and 20. The observed accuracy gains suggest that topic structuring provides additional contextual grounding, which helps compensate for weaker internal representations in lower-resource languages. However, this approach does not yield uniform benefits across all languages. Specifically, Tamil experiences a decline in performance across most of these nuanced topics, indicating that explicit topic awareness does not always align with the model's existing representations or may interfere with its learned knowledge. These findings emphasize the varied impact of LDA-based topic structuring across languages, underscoring the need for language-adaptive methodologies that account for linguistic diversity and dataset-specific factors in hallucination detection.

### 5.6 CASE STUDY E: EFFECT OF TOPIC AWARENESS AND RETRIEVAL CONTEXT IN HALLUCINATION DETECTION

Beyond prompt design, structured contextual information can influence hallucination detection, particularly in multilingual settings where language resources and model training data vary. To examine this, we evaluate the impact of explicit topic awareness (LDA) and retrieval-augmented context (LDA+RAG) on factual claim classification. This analysis considers how topic metadata from LDA and external references from dense passage retrieval (DPR) affect model accuracy across languages.

Hallucination detection accuracy under different evaluation settings is summarized in Table 4. The results indicate that topic structuring improves accuracy for lower-resource languages, while retrieval-based evaluation has mixed effects. Arabic and Amharic see the highest gains with LDA, increasing accuracy by 14.23% and 6.71%, respectively, suggesting that structured topic guidance helps compensate for weaker internal representations. In contrast, English and Chinese experience slight declines, implying that explicit topic structuring may interfere with the model's existing topic awareness.

Retrieval-based evaluation (LDA+RAG) demonstrates varied results. Arabic benefits the most, with accuracy increasing by 10.19%, followed by Amharic at 5.96%, suggesting that retrieved evidence aids languages with limited web resources. However, retrieval slightly reduces accuracy in Chinese and Japanese, possibly due to conflicts with the model's internal knowledge.

The evaluation also highlights language-specific challenges. Some languages, as Tamil and Thai, exhibit minor or negative shifts across different settings, indicating that tokenization, morphology, and dataset biases affect structured context processing. While topic structuring and retrieval augmentation benefit certain languages, they are not universally effective. The findings emphasize the adaptive evaluation frameworks that account for linguistic diversity and dataset properties.

## 6    CONCLUSION

We introduce *Poly-FEVER*, a multilingual benchmark with 77,973 claims across 11 languages, designed to evaluate hallucination detection in LLMs. By benchmarking models like ChatGPT and LLaMA, we reveal significant performance disparities, where accuracy in high-resource languages like English correlates with greater web presence, while lower-resource languages like Amharic lag behind. Our analysis shows that topic structuring (LDA) and retrieval-augmentation (RAG) can improve performance for lower-resource languages, but these benefits are not universal and can conflict with the strong internal knowledge of models in high-resource contexts. *Poly-FEVER* enables researchers to disentangle model, prompt, and resource effects, providing a critical tool for developing more robust and equitable language-adaptive mitigation strategies.

## 7 ETHICAL CONSIDERATIONS

The construction of *Poly-FEVER* raises ethical considerations around fairness, inclusivity, data provenance, and environmental impact. *Poly-FEVER* is designed to reduce linguistic imbalance in hallucination detection research by incorporating 11 languages spanning high-, medium-, and low-resource groups. This inclusivity aims to counteract the overrepresentation of English and a few widely studied languages, promoting more equitable evaluation of large language models. At the same time, uneven performance across languages may reveal systemic inequities in current LLM development.

Biases in data collection, annotation, and curation can skew model predictions, leading to uneven performance across demographic and linguistic groups. Evaluating these biases in factual verification tasks is necessary to avoid reinforcing misinformation or amplifying systemic disparities.

All data sources in *Poly-FEVER* are derived from publicly available fact-checking datasets, ensuring transparency and compliance with ethical guidelines. No private or personally identifiable information is included.

Finally, addressing fairness, bias, and accessibility requires collaboration across disciplines, including linguistics, ethics, and policy. By releasing *Poly-FEVER*, we aim to support transparent, reproducible, and equitable research on multilingual hallucination detection.

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

APPENDIX

# A   *Poly-FEVER* BENCHMARK DETAILS

To support reproducibility in line with ICLR standards, we provide the following:

- **Dataset availability:** *Poly-FEVER* is derived entirely from public benchmarks (FEVER, Climate-FEVER, SciFact). No private or user data are included. Upon acceptance, the dataset will be released on HuggingFace Datasets (anonymized link during review).

- **Code availability:** All preprocessing, evaluation scripts, prompts, and experiment logs will be released in a public GitHub repository. This includes random seeds, pinned requirements, and command-line scripts.

- **Evaluation:** All models are tested with fixed random seeds. Deterministic inference is enforced where supported.

- **Cost:** Multilingual translation was conducted using Google Cloud Translation, costing $2,644 for 10 non-English languages.

## A.1   DATASET STATISTICS

Character length varies systematically across languages even for identical claims. The main drivers are script density (e.g., logographic vs. alphabetic/abjad), morphology (agglutinative vs. isolating), and orthographic conventions (spacing and punctuation). As a result, the same semantics tend to compress in Chinese/Japanese and expand in languages with richer inflection or compounding (e.g., Tamil, Amharic), consistent with the ranges in Table 5.

| Language | Chinese | Hindi | Arabic | Bengali | Japanese | Korean | Tamil | Thai | Georgian | Amharic |
|---|---|---|---|---|---|---|---|---|---|---|
| **Min (chars)** | 6 | 6 | 14 | 12 | 11 | 10 | 16 | 12 | 16 | 2 |
| **Max (chars)** | 0.6k | 1.8k | 1.9k | 1.9k | 0.8k | 0.9k | 2.2k | 1.7k | 2.0k | 1.2k |

Table 5: Character-length ranges (min and max) of claim strings by language in the *Poly-FEVER* metadata.

## A.2   EXAMPLE RECORD

| Field | Value |
|---|---|
| ID | 1 |
| Label | TRUE |
| Topic Distribution (top-5) | 16, 21, 0, 1, 2 |
| en | Aramais Yepiskoposyan played for FC Ararat Yerevan, an Armenian football club based in Yerevan during 1986 to 1991. |
| zh-CN | Aramais Yepiskoposyan 于 1986 年至 1991 年间效力于 FC Ararat Yerevan，这是一家位于埃里温的亚美尼亚足球俱乐部。 |
| hi | अरामाइस येपिस्कोपोसियन 1986 से 1991 के दौरान येरेवन में स्थित अर्मेनियाई फ़ुटबॉल क्लब एफसी अरारत येरेवन के लिए खेले। |
| ar | لعب أراميس يبيسكوبوسيان لصالح نادي أرارات يريفان، وهو نادي كرة قدم أرمني مقره في يريفان خلال الفترة من 1986 إلى 1991. |
| bn | Aramais Yepiskoposyan FC Ararat Yerevan, 1986 থেকে 1991 সাল পর্যন্ত ইয়েরেভানে অবস্থিত একটি আর্মেনিয়ান ফুটবল ক্লাবের হয়ে খেলেছিলেন। |
| ja | アラマイス・イェピスコポシアンは、1986年から1991年までエレバンに本拠を置くアルメニアのサッカークラブ、FCアララト・エレバンでプレーした。 |
| ko | Aramais Yepiskoposyan은 1986년부터 1991년까지 예레반을 연고로 하는 아르메니아 축구 클럽인 FC Ararat Yerevan에서 뛰었습니다. |
| ta | Aramais Yepiskoposyan 1986 முதல் 1991 வரை யெரெவனில் உள்ள ஆர்மீனிய கால்பந்து கிளப்பான FC அரரத் யெரெவனுக்காக விளையாடினார். |
| th | Aramais Yepiskoposyan เล่นให้กับ FC Ararat Yerevan สโมสรฟุตบอลอาร์เมเนียที่ตั้งอยู่ในเยเรวานระหว่างปี 1986 ถึง 1991 |
| ka | არამაის იეპისკოპოსიანი თამაშობდა FC Ararat Yerevan-ში, სომხურ საფეხბურთო კლუბში, რომელიც დაფუძნებულია ერევანში 1986-1991 წლებში. |
| am | አራማይስ ዬፒስኮፖስያን ከ1986 እስከ 1991 በፈረንጆች አቆጣጠር ለአርሜኒያ እግር ኳስ ክለብ ለአርኤፍ አራራት ዬሬቫን ተጫውቷል። |

Figure 6: Example record from *Poly-FEVER*. Claims are aligned across 11 languages with a shared truth label and top-5 LDA topics. Cross-script variation highlights differences in length and tokenization.

Each entry in *Poly-FEVER* is a *single claim* aligned across 11 languages with a *shared* binary veracity label and lightweight topic metadata. Concretely, a record contains: ID, Claim (parallel text in 11 languages), Label (true/false), and the top-5 TopicIDs from LDA. The fields are designed to be minimal yet sufficient for (i) multilingual fact verification, (ii) hallucination analysis, and (iii) topic-aware diagnostics. All fields are required and stored in UTF-8 NFC. Language tags follow BCP-47; we use en, zh-CN, hi, ar, bn, ja, ko, ta, th, ka, am. The Label is *identical* for all languages of the same record; translations are intended to be logically equivalent (Section 4.3 discusses extrinsic conditioning). TopicIDs are the indices of the five most probable LDA topics for the *English* surface form (Section A.3); they provide a shared topical anchor across languages.

### A.3 TOPIC MODELING VIA LDA

LDA models each claim as a mixture of topics, with each topic represented by a distribution of words. The model assumes that claims share underlying themes and assigns probability distributions to capture these relationships. Each claim consists of weighted topic assignments that indicate its alignment with different themes. The model learns these distributions by adjusting word probabilities to match patterns observed in the dataset. We evaluate topic classifications from 0 to 50 and determine that 22 topics (Table 6) achieve the highest coherence scores on *Poly-FEVER*.

To prepare claims for topic modeling, we apply a preprocessing pipeline that standardizes text for analysis. The pipeline converts all text to lowercase, tokenizes words, removes typos, eliminates stopwords, and applies stemming and lemmatization. This process ensures that LDA captures relevant word distributions without noise from text inconsistencies.

After preprocessing, we construct a Gensim Dictionary (Rehurek & Sojka, 2011) and transform the corpus into a Bag-of-Words (BoW) representation. We apply Term Frequency-Inverse Document Frequency (TF-IDF) weighting to refine feature representation. The LDA model runs for 200 iterations to identify topic distributions and optimize coherence scores. The model assigns probabilities to claims based on topic composition, revealing structures that influence hallucination patterns in multilingual verification.

| ID | Topic | ID | Topic | ID | Topic |
|----|-------|----|-------|----|-------|
| 0 | Politics | 8 | Warfare History | 16 | Soccer |
| 1 | Music | 9 | Military Services | 17 | Biology/Taxonomy |
| 2 | Sports | 10 | Biology/Genetics | 18 | Architecture/History |
| 3 | Aviation | 11 | Equestrian | 19 | Business/Finance |
| 4 | Competitive Sports | 12 | Architecture/Construction | 20 | Film/Television |
| 5 | American Football | 13 | Entertainment | 21 | Transportation/Infrastructure |
| 6 | Photography/Arts | 14 | Automotive Racing | | |
| 7 | Demographics | 15 | Education | | |

Table 6: Number-topic references table for LDA topic distribution on 22 topics. This table presents the topic assignments derived from LDA applied to the *Poly-FEVER* dataset. These topic classifications serve as a basis for evaluating the impact of topic awareness.

## B CAUSES EXPLORATION

### B.1 EFFECT OF PROMPTS

We provide verbatim text for the three prompt families (General, Language-wise, and Classification) and their translations for all 11 languages. For completeness, we also include the minor variants used in ablations.

We designed both a language-wise prompt and a classification prompt (each with 11 translated versions) to guide LLMs toward producing either *true* or *false* outputs for fact-checking tasks. In early trials, lengthy prompts filled with additional context often confused the models, leading them to ignore specific instructions and produce irrelevant explanations. To mitigate this issue, we refined the prompts through several iterations, prioritizing clarity and brevity so that the task constraints were explicit without adding unnecessary complexity. Ultimately, we developed a concise general prompt to classify factual claims as *true* or *false* based on truthfulness, thereby reducing irrelevant explanations and improving the accuracy of LLM responses.

> **General Prompt Baseline Evaluation**
>
> Label the following claim with truthfulness into: *true* and *false*. No other characters, no punctuation, all lower size.

Nevertheless, LLMs exhibit overenthusiasm by providing detailed explanations or references, straying from the intended binary classification of each claim as *true* or *false*. To counter this, we restructured the language-wise prompt into a three-sentence directive outlining the overall task definition,

the precise answer format, and forbidden responses. We found that performance dropped when the prompt and claim were in different languages, so we translated the standardized prompt into the same 11 languages as the claims. This alignment between prompt language and claim language establishes to curb irrelevant model outputs and ensures consistency in binary fact-checking responses.

> **Language-wise Prompt Language-wise Prompt for Multilingual Evaluation**
>
> Assess the truthfulness of the user's claim and provide a response. Use *true* to indicate that the claim is true, *false* to indicate that it is false. Your response should only consist of *true* or *false*, without any additional characters or punctuation.

Although LLMs are directed to evaluate claims as *true* or *false* without contextual analysis, their responses are integrated sentence structure. Although LLMs are directed to output only *true* or *false*, their responses often appear within broader sentence structures. To improve classification accuracy, we moved beyond simple pattern matching and designed a classification prompt that captures how LLMs express these labels in context.

> **Classification Prompt Free-form Output Mapping**
>
> Classify the input as *true* or *false* based solely on the indicative words or phrases within it. Assign *true* if the input contains affirming terms such as *Correct*, *TRUE*, *really*, or *the truth*. Assign *false* if it contains negating or contradictory terms such as *Fake*, *False*, or any phrasing that signals correction or contradiction. Respond with only *true* or *false* for the input, without any additional text, characters, or punctuation.

By filtering responses through this structured classification prompt, we ensure that outputs conform to a strict binary format. This approach standardizes hallucination detection across multiple languages, preventing inconsistencies in model interpretation.

### B.2 EXTERNAL KNOWLEDGE RETRIEVAL

We employed the RAG technique, leveraging its state-of-the-art capabilities to bolster the accuracy and relevance of responses produced by LLMs within our fact-checking framework. This system applies the Dense Passage Retrieval (DPR) mechanism (Karpukhin et al., 2020), which utilizes embeddings for document retrieval. RAG revolutionizes NLP by amalgamating generative models with an external knowledge retrieval component, enabling dynamic access to a vast corpus of information. This external augmentation enhances the model's internal knowledge base with pertinent external data during generation.

For external retrieval, we employ the wiki_dpr dataset, an extensive collection of 21 million Wikipedia passages, each adorned with DPR embeddings. These documents are segmented into 100-word, non-overlapping text blocks, optimizing the dataset for precise analysis and the evaluation of DPR's retrieval efficacy.

Leveraging the DPR-embedded Wikipedia corpus, we construct a Facebook AI Similarity Search (FAISS) based dense index over **21 million passages**, enabling sub-second nearest-neighbor retrieval. Each claim is encoded with DPR's Question Encoder, matched against the index, and augmented with the top-5 semantically aligned passages. This augmentation equips the LLM with contextualized evidence that can mitigate *extrinsic hallucinations* by grounding generations in verifiable text. This process retrieves documents that are semantically aligned with the claim, giving the LLM an input context that combines the claim, task constraints, and supporting evidence—helping ground predictions and reduce reliance on uneven internal knowledge across languages.

## C  EVALUATION

### C.1 EVALUATION SETUP (EXTENDED)

Our selection of ChatGPT, LLaMA-2, and LLaMA-3.1 was driven by their extensive language support and significant influence in the AI field. Although LLaMA-2 is primarily designed for English, it includes 27 other languages (Touvron et al., 2023), prompting us to examine its non-English

hallucination detection. Initial evaluations (see Table 2: columns 4, 5, 6) revealed its multilingual limitations, as its performance approximated random guessing. Therefore, we extended our evaluation to LLaMA-3.1, which introduces improvements in multilingual capabilities and factual consistency. This comparison allows us to assess whether the latest iteration reduces hallucinations in non-English settings.

We conduct evaluations of ChatGPT 3.5 Turbo, LLaMA-2 (7B, 13B, and 70B), and LLaMA-3.1 8B on a server equipped with a 12-core CPU and dual NVIDIA A5500 GPUs, each with 24GB of memory. For the evaluation of the larger LLaMA-2 70B model, we use a more powerful machine comprising 2 NUMA nodes. Each of these nodes features a 20-core CPU and 4 NVIDIA V100 GPUs with 32GB of memory, interconnected via NVLink to ensure fast and efficient data transfer. The LLaMA-3.1 8B model is evaluated on the same hardware as LLaMA-2 7B and 13B, ensuring a consistent computational environment for comparison. All models are evaluated using the PyTorch deep learning framework, version 2.0.1. Furthermore, the temperature of LLaMA-2 is set to 0, which is intended to yield deterministic results.

## C.2 TEMPERATURE SENSITIVITY SWEEP

To further evaluate the performance of LLaMA 3.1 8B model, we conducted experiments on the TACC Lonestar6 system equipped with three NVIDIA A100 GPUs, each with 40GB of HBM2 memory, alongside a 64-Core processor with 128 total cores and 256GB RAM. The model was evaluated across temperature values ranging from 0.1 to 2.0. The results, as demonstrated in Table 7, indicate that languages such as English exhibit improved performance with higher temperatures, peaking around 2.0, while others, like Chinese and Amharic, decline in performance as the temperature increases. Languages such as Japanese and Bengali maintain relatively stable performance across temperatures, while others like Tamil and Korean exhibit fluctuations. Overall, the optimal temperature setting varies by language, with some benefiting from higher temperatures and others being more stable at lower ones, indicating that temperature tuning is critical for achieving the best results across languages.

| Lang. | 0.1 | 0.2 | 0.3 | 0.4 | 0.5 | 0.6 | 0.7 | 0.8 | 0.9 | 1.0 | 1.1 | 1.2 | 1.3 | 1.4 | 1.5 | 1.6 | 1.7 | 1.8 | 1.9 | 2.0 |
|---|---|---|---|---|---|---|---|---|---|---|---|---|---|---|---|---|---|---|---|---|
| en | 47.72 | 47.71 | 47.74 | 47.8 | 47.78 | 47.89 | 48.04 | 47.97 | 48.06 | 48.25 | 48.39 | 48.52 | 48.47 | 48.78 | 48.79 | 48.83 | 49.06 | 49.07 | **49.26** | 49.17 |
| zh-CN | **55.79** | 55.66 | 55.53 | 55.43 | 55.31 | 55.28 | 54.88 | 54.67 | 54.56 | 54.52 | 54.28 | 54.00 | 54.02 | 54.02 | 53.85 | 53.52 | 53.47 | 53.24 | 52.95 | |
| hi | 46.72 | 46.70 | 46.73 | 46.98 | 46.96 | 47.22 | 47.58 | 47.64 | 47.65 | 47.86 | 48.00 | 48.23 | 48.23 | 48.69 | 48.68 | 48.54 | 48.51 | 48.43 | 48.63 | **48.84** |
| ar | 48.41 | 48.38 | 48.49 | 48.64 | 48.66 | 49.16 | 49.06 | 49.18 | 49.28 | 49.23 | 49.62 | 49.60 | 49.70 | 49.62 | 49.92 | 49.66 | 49.97 | **50.07** | 49.99 | 49.86 |
| bn | 42.71 | 42.85 | 43.21 | 43.52 | 44.03 | 44.58 | 45.28 | 45.40 | 45.47 | 46.28 | 46.69 | 46.99 | 47.27 | 47.20 | 47.04 | **47.56** | 47.52 | 47.46 | 47.53 | |
| ja | 56.08 | **56.10** | 55.96 | 55.99 | 55.96 | 55.95 | 55.61 | 55.62 | 55.54 | 55.40 | 55.37 | 55.25 | 55.23 | 55.23 | 55.05 | 55.16 | 54.70 | 54.65 | 54.41 | 54.29 |
| ko | 52.97 | 52.86 | **52.99** | 52.78 | 52.82 | 52.82 | 52.78 | 52.78 | 52.73 | 52.58 | 52.63 | 52.22 | 52.41 | 52.52 | 52.18 | 51.96 | 52.28 | 51.97 | 51.88 | 51.84 |
| ta | 55.87 | **55.91** | 55.71 | 55.25 | 54.94 | 54.63 | 53.87 | 53.72 | 53.72 | 53.09 | 53.00 | 52.45 | 52.34 | 52.06 | 51.76 | 51.62 | 51.43 | 50.70 | 50.49 | 50.35 |
| th | 49.61 | 49.76 | 49.92 | 49.86 | 49.89 | 50.04 | 50.22 | 50.15 | 50.22 | 50.48 | 50.45 | 50.54 | 50.77 | 50.61 | 50.77 | 50.81 | 50.66 | **50.93** | 50.58 | 50.46 |
| ka | 47.86 | 47.94 | 48.19 | 48.13 | 48.35 | 48.40 | 48.59 | 48.66 | 48.64 | 48.79 | 48.91 | **49.26** | 49.09 | 49.04 | 49.16 | 49.26 | 48.95 | 48.86 | 48.65 | 48.61 |
| am | 53.75 | **53.43** | 52.69 | 51.80 | 51.26 | 50.43 | 49.76 | 49.32 | 49.06 | 47.73 | 47.12 | 47.02 | 46.52 | 46.16 | 46.39 | 46.35 | 46.22 | 46.09 | 46.20 | 46.48 |

Table 7: Performance of LLaMA-3.1 8B across temperature values from 0.1 to 2.0 for 11 languages. The highest accuracy for each language is highlighted in bold, while the lowest is underlined. The results provide insights into how different temperature settings influence multilingual fact verification and hallucination patterns across diverse linguistic contexts.

## C.3 HALLUCINATION ON MULTILINGUAL FACT-CHECKING TASK

As shown in Table 8, for ChatGPT 3.5, it is observed that the model demonstrates great stability across all topic fields (general, climate, and science facts), with English consistently showing the highest judgment accuracy and Arabic showing the lowest judgment accuracy. The performance gap between the highest and lowest percentages for the three datasets are 23.11%, 38.22%, and 35.06%, indicating the greatest variability in the Climate-FEVER and the least in FEVER.

LLaMA-2 (7B, 13B, and 70B) results are illustrated in Table 9, 10, and 11 respectively. As the size of the LLaMA2 models increases, a noticeable bias towards different languages in LLMs becomes apparent. In the case of LLaMA-2 7B and 13B, Amharic exhibits the poorest performance in general fields, whereas Bengali demonstrates the weakest performance in categorizing science claims. LLaMA2 70B displays significant variation in performance across different languages, particularly in terms of the lowest accuracy. In general and science topics, Tamil exhibits remarkably low estimation rates, registering only 15.00% and 10.17%, respectively.

| Lang. | FEVER | Climate-FEVER | SciFact |
|---|---|---|---|
| **en** | **65.89%** | **74.29%** | **71.57%** |
| **zh-CN** | 58.25% | 55.12% | 54.83% |
| **hi** | 52.90% | 41.79% | 42.14% |
| **ar** | 45.48% | 36.07% | 36.51% |
| **bn** | 55.65% | 41.19% | 43.15% |
| **ja** | 55.89% | 55.00% | 55.56% |
| **ko** | 57.29% | 50.60% | 54.11% |
| **ta** | 55.67% | 55.95% | 43.58% |
| **th** | 56.82% | 46.31% | 44.44% |
| **ka** | 57.37% | 50.36% | 49.93% |
| **am** | 47.53% | 39.52% | 37.95% |

Table 8: Accuracy of hallucination detection by ChatGPT 3.5 on *Poly-FEVER*.

| Lang. | FEVER | Climate-FEVER | SciFact |
|---|---|---|---|
| **en** | **63.35%** | **77.70%** | **70.71%** |
| **zh-CN** | 58.29% | 58.59% | 62.46% |
| **hi** | 48.59% | 41.92% | 40.33% |
| **ar** | 50.55% | 46.53% | 51.52% |
| **bn** | 49.05% | 46.31% | 36.04% |
| **ja** | 58.24% | 61.23% | 64.54% |
| **ko** | 56.67% | 52.36% | 63.87% |
| **ta** | 49.54% | 46.56% | 35.03% |
| **th** | 53.90% | 44.73% | 48.49% |
| **ka** | 47.39% | 43.00% | 37.31% |
| **am** | 43.42% | 48.15% | 45.26% |

Table 9: Accuracy of hallucination detection by LLaMA-2 7B on *Poly-FEVER*.

| Lang. | FEVER | Climate-FEVER | SciFact |
|---|---|---|---|
| **en** | **64.27%** | **72.00%** | **72.44%** |
| **zh-CN** | 59.88% | 62.82% | 66.57% |
| **hi** | 54.33% | 56.06% | 57.33% |
| **ar** | 54.97% | 56.77% | 61.41% |
| **bn** | 52.30% | 51.01% | 52.25% |
| **ja** | 59.57% | 66.46% | 67.65% |
| **ko** | 58.74% | 62.97% | 67.36% |
| **ta** | 50.86% | 50.38% | 61.02% |
| **th** | 53.46% | 45.57% | 58.79% |
| **ka** | 53.45% | 55.00% | 62.69% |
| **am** | 48.64% | 44.44% | 53.68% |

Table 10: Accuracy of hallucination detection by LLaMA-2 13B on *Poly-FEVER*.

| Lang. | FEVER | Climate-FEVER | SciFact |
|---|---|---|---|
| **en** | **64.56%** | **78.42%** | **75.32%** |
| **zh-CN** | 38.75% | 49.72% | 42.08% |
| **hi** | 45.68% | 45.96% | 33.67% |
| **ar** | 32.97% | 30.03% | 31.92% |
| **bn** | 33.87% | 19.46% | 18.02% |
| **ja** | 41.37% | 51.27% | 42.77% |
| **ko** | 46.06% | 47.88% | 51.31% |
| **ta** | 19.47% | 19.85% | 10.17% |
| **th** | 48.09% | 31.22% | 30.65% |
| **ka** | 46.15% | 40.00% | 33.03% |
| **am** | 26.34% | 11.11% | 22.11% |

Table 11: Accuracy of hallucination detection by LLaMA-2 70B on *Poly-FEVER* data.