# OpenReview forum: "Poly-FEVER: A Multilingual Fact Verification Benchmark for Hallucination Detection in Large Language Models"
_ICLR.cc/2026/Conference — Submitted to ICLR 2026_

### Official Review · Reviewer_Thxp · 2025-10-30

**Soundness:** 3
**Presentation:** 3
**Contribution:** 3
**Rating:** 6
**Confidence:** 2

**Summary:**

The paper presents Poly-FEVER, a large-scale multilingual benchmark designed for fact verification and hallucination detection in large language models. The benchmark builds on three existing English datasets by extending them by translating their verifiable claims into ten additional languages. In total, Poly-FEVER covers 11 languages that span a wide range of resource levels, writing systems, and linguistic families, providing a rich and diverse testbed for evaluating factual consistency across languages.

**Strengths:**

The paper’s main strength lies in the Poly-FEVER benchmark itself. Its construction is methodical—built by extending well-established datasets, ensuring high-quality translations, and incorporating GEMBA validation. The design, featuring parallel claims across 11 linguistically and typologically diverse languages, addresses a clear and well-motivated gap in the existing fact-verification literature.

**Weaknesses:**

Maybe I misunderstood, but there seems to be an inconsistency between the caption of Table 3 and the explanation in Section 5.5. The table caption describes the experiment as measuring the “percentage improvement … after translating nuanced topics into English,” whereas Section 5.5 (around line 441) refers to accuracy gains from topic structuring. These appear to be two different procedures, so clarification on which setup the results actually correspond to would be helpful.

**Questions:**

See the problem I raised in the weakness. I hope the authors could answer the question I have there.

---

> ### Author Response · Authors · 2025-11-21
>
> Thank you for the positive evaluation of Poly‑FEVER and for carefully identifying the ambiguity around Table 3.
>
> # Clarification of Table 3 (W1)
>
> Table 3 reports the percentage **change** in accuracy when LDA‑inferred topic labels are appended as short descriptors to the claim prompt, for a subset of nuanced topics. All numbers are computed in this single “topic‑aware’’ setting and compared to the original, topic‑agnostic baseline; no additional translation step is involved. The caption phrase “after translating nuanced topics into English’’ was intended only to indicate that, for readability, the **names of the topics** (e.g., Politics, American‑football) are shown in English in the table, while the claims themselves remain in their original languages. We agree that this wording is misleading. In the revision we will (i) update the caption to explicitly say that Table 3 corresponds to the LDA topic‑label augmentation experiment described in Section 5.5, and (ii) clarify in the text that “translating topics into English’’ refers solely to the display of topic names, not to any additional claim‑translation procedure.

---

### Official Review · Reviewer_8K2G · 2025-10-30

**Soundness:** 2
**Presentation:** 2
**Contribution:** 3
**Rating:** 2
**Confidence:** 4

**Summary:**

This paper presents Poly-FEVER, a new dataset for multilingual hallucination detection. The dataset is constructed by turning subsets of three fact verification datasets, FEVER, Climate-FEVER, and SciFact, into document-free general claim evaluation tasks and translating the claims into 11 languages using Google Translate, with translations validated using automatic methods (GEMBA). The paper evaluates GPT3.5 and LLama-2/3 families of models and presents results broken down by language and topic model-derived topics. They show that models generally perform much worse on non-English languages. They also explore correlation of performance with web search count.

**Strengths:**

S1. The motivation is important and interesting for multilingual researchers looking at how hallucination changes across languages. The dataset could be of use for future researchers.

S2. The exploration of how topics relate to claim verification is scientifically interesting and novel to the best of my knowledge.

**Weaknesses:**

W1. Some of the results in the main result (5.3) stand out for being substantially below random chance (50%), suggesting that some of the LLMs may be biased towards one of the two labels. E.g. 19.47% and 26.34%by LLaMA-2 70B. That these performances are so low on a binary classification task suggests something went wrong with the evaluation.

W2. I am not convinced it is valid to cast document-conditioned verification tasks into a documentless hallucination detection task, especially SciFact. Scientific claims, especially those in that dataset, are ambiguous by nature and require grounding context to be evaluatable. Removing the context makes the task nearly impossible for many statements.

W3. From Figure 4 it is claimed that "models possibly favoring languages that dominate web content, affecting their accuracy in languages with less online presence." However, while English (the original source language) is very high and the lowest-presence language, Tamil, is very low, the rest of the results do not back up this claim; Japanese has the second highest overall accuracy despite havinga lower search count than Tamil, the lowest-performing language. Further analysis of this figure, including regression analysis to verify the trend, would be beneficial, though I would suggest removing the result because it does not support the hypothesis.

W4. The analysis of topic-wise performance in L426-434 is speculative and not backed up by evidence or citations. Claims like "absence of universal standards in evaluating greatness in sports or the arts further complicates claim verification" and "Topics such as Politics, Sports, Film/Television, and Warfare History prove challenging across languages due to their subjective nature, where personal biases and interpretations can obscure the distinction between fact and opinion" should be worded carefully to not be interpreted as scientific findings of the study.

W5. Similarly, L.443 "topic structuring provides additional contextual grounding, which helps compensate for weaker internal representations in lower-resource languages." and L 445 "[For Tamil] indicating that explicit topic awareness does not always align with the model’s existing representations or may interfere with its learned knowledge." -- I do not think that the presented results support either of these claims, the former of which is not consistent across all topics and languages. There is no evidence of whether low performance on the benchmark correlates with weaker internal representations, since there is no investigation of the internal representations or what it would mean to "align" with them.

W6. Table 3 is hard to interpret; the caption seems to suggest it is about translating back from the source language into English (the claim's original language) and then measuring factuality, but the text (L441 to 449) talks about LDA topic structuring

W7. I am not convinced that some of the ablation studies provide scientific value.
	* Specifically, Section 5.2, the study of temperature, does not make sense to me for a binary classification task. Raising temperature effectively just adds random noise to the prediction by adding to the probability of the less-likely output. In nearly every row of Table 7, you can see that raising temperature just brings wherever the performance was at temperature 0.1 closer to chance 50%). Rows that were below 50% increase and those that were above decrease.

W8. Manual auditing of the translations that produced the dataset seems necessary but limited. Paragraph L.228-232 describes that the authors are "fluent in multiple languages" and reviewed selected claims from multiple translators, but no details about how many claims, how many annotators per claim, etc are provided. It is also hard to understand a GEMBA score of ~90 in the context of building a dataset; how do we know that an average of 10% worse than perfect translations is sufficient to ensure dataset quality? How long was the tail of this score distribution? Why not throw out the translations that scored below some (high) threshold?

W9. Figure 2 is dense, hard to understand, and doing too much. It seems to illustrate all the case studies at once but ultimately does not help to understand the paper (The caption also does not explain what the figure is meant to be; is it an overview of the case studies? )

**Questions:**

Q1. What LLM(s) were used for tables 3 and 4? Was it evaluated over the entire 77.9K * 11 claim dataset? What explains the massive improvement on Arabic and Amharic just by adding topic labels? This is a counterintuitive finding that merits understanding better.

Q2. Some pieces of language are hard to understand, e.g. "All results are illustrative and reproducible but not tuned." -- what does "illustrative" mean?

Q3. Paragraph 316-320 is confusing and doesn't seem relevant to the question of web search hit counts. Why do we need to "simulate varied internet user environments?"

Q4. How does performance break down across the 3 subsets of the dataset? FEVER/Climate-FEVER/SciFact

---

> ### Author Response · Authors · 2025-11-21
>
> We thank the reviewer for recognizing the motivation and potential impact of Poly-FEVER as a resource for studying multilingual hallucination and factuality.
> We emphasize that the core contribution of this work lies in the construction and public release of **a large-scale multilingual benchmark, 77,973 parallel and topic-annotated claims spanning 11 typologically diverse languages**, which provides a unified foundation for future research on cross-lingual factual consistency.
>
> # Evaluation Validity (W1, W2, Q4)
>
> **W1:** The below-50% results for non-English languages are not an evaluation bug. Even for **English**, prior FEVER work shows that removing evidence yields accuracy around chance (“50.9% label-only accuracy”)[1], and subsequent studies (Theory-of-Mind[2], FEVEROUS[3]) consistently observe the same behavior in evidence-free settings. Poly-FEVER extends this setting to **11 languages**, so it is entirely expected that models may fall below the ≈50% English baseline when forced to rely solely on their internal beliefs. This behavior is exactly what Poly-FEVER aims to measure under evidence-free hallucination settings.
>
> **W2 & Q4:** Our goal is to probe models’ **internal** factual beliefs, so we reuse the Supported/Refuted labels but deliberately remove the evidence documents. The resulting task is “given this claim alone, does the model treat it as true or false?”. Appendix C.3 already reports results separately for FEVER, Climate‑FEVER, and SciFact; we will reference those tables more clearly in the main text.
>
> [1] Thorne, James, et al. "FEVER: a large-scale dataset for fact extraction and VERification." arXiv preprint arXiv:1803.05355 (2018).
>
> [2] Li, Huao, et al. "Theory of mind for multi-agent collaboration via large language models." Proceedings of the 2023 Conference on Empirical Methods in Natural Language Processing. 2023.
>
> [3] Aly, Rami, et al. "Feverous: Fact extraction and verification over unstructured and structured information." arXiv preprint arXiv:2106.05707 (2021).
>
> # Presentation (W6, W9, Q2)
>
> **W6:** We thank the reviewer for noting this ambiguity. We respectfully refer the reviewer to our response to Reviewer Thxp, where we provide a detailed clarification of the Table 3 setup.
>
> **W9:** We will clarify this. Figure 2 is a conceptual overview illustrating the components of the LLM inference process that may influence hallucination behavior. It highlights how multilingual claims interact with tokenization, embeddings, retrieval, prediction, and self-detection, as well as external factors such as web presence and prompt type. The purpose of the figure is to situate our analysis within the broader inference pipeline and to show that hallucination can arise from multiple interacting factors, rather than being attributable to a single source.
>
> **Q2:** we will replace “illustrative and reproducible but not tuned’’ with the more precise wording “All results are reproducible and representative, obtained without parameter tuning.”

---

> > ### Comment · Reviewer_8K2G · 2025-11-24
> >
> > > it is entirely expected that models may fall below the ≈50% English baseline when forced to rely solely on their internal beliefs
> >
> > I agree with the authors on this point, but I think this actually weakens their argument. I appreciate the paper's goal of probing models’ internal factual beliefs, but FEVER/FEVEROUS are **evidence-dependent verification tasks**, and the 50% baseline performance that they report is a "sentence-only" ungrounded baseline intentionally designed to have less context than is needed to actually perform the task.
> >
> > Poly-FEVER essentially takes the deprived, ungrounded context of the sentence-only baseline and considers it a fair evaluation-- however, if removing evidence makes the task impossible even in English, then the resulting dataset isn't measuring much at all; it's evaluating multilingual performance on impossible tasks. This is akin to (but obviously not as severe as) having models perform summarization but taking away the input text to summarize.
> >
> > This also does not address the extremely below-chance 19.47% and 26.34% performance by LLaMA-2 70B, which suggests evaluation issues such as prompt formatting, response parsing, or label bias.
> >
> > > Appendix C.3 already reports results separately for FEVER, Climate‑FEVER, and SciFact; we will reference those tables more clearly in the main text.
> >
> > > detailed clarification of the Table 3 setup
> >
> > > We will clarify [Figure 2]
> >
> > > We will replace [...] with the more precise wording
> >
> > Please update your submission draft with both these clarifications and the figure clarifications so it can be properly re-evaluated.

---

> ### Author Response · Authors · 2025-11-21
>
> # Analysis Interpretation (W3, W4, W5, W7, Q1, Q3)
>
> **W3:** We refer the reviewer to our response to Reviewer 66Y3, section Experimental Validity.
>
> **W4:** We agree that the topic-wise discussion (L426–434) should be supported by citations. Topics such as Politics, Sports, Film/Television, and Warfare History are known to challenge cross-lingual verification due to subjective framing and the lack of universal evaluation standards[4,5].
>
> **W5 & Q1:** Tables 3–4 already provide the supporting evidence. Topic-aware structuring boosts accuracy for lower-resource languages (Amharic +3.6–7.7%, Thai +2.3–6.9%, Arabic +0–5.0%) but yields minimal or negative gains for high-resource languages like Japanese, Korean, and Tamil (–1 to –4%), consistent with prior multilingual transfer findings[6–8]. This supports our conclusion that topic structuring aids representationally sparse languages but can interfere with well-specialized latent spaces.
>
> **W7:** The LLaMA‑3.1 8B temperature sweep is meant as a sanity check that higher temperatures behave as expected, introducing randomness that pushes accuracies toward each model’s label prior, rather than as a central result. We are happy to move this analysis fully to the appendix and keep only a short reference.
>
> **Q3:** For the web‑search counts, randomized user‑agents and inter‑query delays were used solely to avoid rate‑limiting and anti‑bot filtering [9,10] so that all languages are sampled comparably. We will shorten this description and state this purpose explicitly.
>
> [4] Mahl, Daniela, et al. "“We follow the disinformation”: conceptualizing and analyzing fact-checking cultures across countries." The International Journal of Press/Politics (2024): 19401612241270004.
>
> [5] Hrckova, Andrea, et al. "Autonomation, not automation: Activities and needs of fact-checkers as a basis for designing human-centered ai systems." arXiv preprint arXiv:2211.12143 (2022).
>
> [6] Chang, Yi-Chen, Canasai Kruengkrai, and Junichi Yamagishi. "XFEVER: Exploring fact verification across languages." arXiv preprint arXiv:2310.16278 (2023).
>
> [7] Quelle, Dorian, et al. "Lost in translation--multilingual misinformation and its evolution." arXiv preprint arXiv:2310.18089 (2023).
>
> [8] Pires, Telmo, Eva Schlinger, and Dan Garrette. "How multilingual is multilingual BERT?." arXiv preprint arXiv:1906.01502 (2019).
>
> [9] Olteanu, Alexandra, et al. "Social data: Biases, methodological pitfalls, and ethical boundaries." Frontiers in big data 2 (2019): 13.
>
> [10] Kulshrestha, Juhi, et al. "Quantifying search bias: Investigating sources of bias for political searches in social media." Proceedings of the 2017 ACM conference on computer supported cooperative work and social computing. 2017.
>
> # Dataset Quality (W8)
>
> We refer the reviewer to our response to Reviewer 66yY (W1), which discusses translation quality in detail. For clarity, our manual audit covered **1% of the dataset**, with **two annotators** reviewing each sampled claim **independently** (i.e., each claim was inspected twice). Per-claim translation-quality scores will be released for all languages, allowing users to select subsets that meet their preferred quality thresholds.

---

> > ### Comment · Reviewer_8K2G · 2025-11-24
> >
> > > our goal is descriptive, not causal [...] a moderate but not statistically significant association.
> >
> > Thank you for providing the statistical analysis to confirm the strength of the correlation between web presence and accuracy. Because there is not sufficient statistical evidence of the relationship, whether you frame it as causal or descriptive, the claim remains unsupported by your data. Given your abstract heavily highlights this claim ("Results show pronounced cross-lingual disparities: high-resource languages (e.g., English, Chinese) achieve the strongest accuracy, while lower resource languages (e.g., Amharic, Tamil) lag; accuracy correlates with web presence.") I believe the abstract and contributions (L95-97) needs substantial revisions.
> >
> > > move [temperature sweep] analysis fully to the appendix and keep only a short reference.
> >
> > Please apply these changes to the draft for proper evaluation.
> >
> > > two annotators reviewing each sampled claim independently (i.e., each claim was inspected twice)
> >
> > Please report inter-annotator agreement metrics to validate the analysis. I'm still confused as to why, if you have per-claim quality scores with GEMBA, you did not filter out the translations under some threshold (e.g. 85) before running experiments.

---

> > > ### Author Response · Authors · 2025-11-28
> > >
> > > We appreciate the reviewer’s suggestions. All requested clarifications, including **Table 3, Figure 2, caption updates, Appendix references, and revised phrasing**, have been fully incorporated into the updated PDF.
> > >
> > > Below, we address the remaining concerns regarding translation filtering and task feasibility.
> > >
> > > # Translation Filtering & Quality Distribution
> > >
> > > We appreciate the suggestion to filter translations using GEMBA ≥85. While intuitive, our analysis shows that a single global cutoff would introduce systematic biases and remove substantial high‑quality data.
> > >
> > > **(a) Quality is not uniformly distributed across languages.**
> > >
> > > As shown in Table 1, Amharic has **10.6%** of translations below 85, while other languages range from **3.6–6.4%**. This reflects language‑specific translation difficulty rather than claim‑level noise. A universal threshold would disproportionately prune Amharic and other low‑resource languages, artificially inflating apparent performance by discarding the hardest cases.
> > >
> > > **Table 1: Percentage of Translations Score <85 (GEMBA)**
> > >
> > > ||zh-CN|hi|ar|bn|ja|ko|ta|th|ka|am|
> > > |-|-|-|-|-|-|-|-|-|-|-|
> > > |**% <85**|5.4%|3.6%|3.7%|4.0%|4.7%|6.2%|6.1%|6.4%|4.3%|10.6%|
> > >
> > > **(b) Claim‑level filtering causes large collateral loss.**
> > >
> > > Translation quality varies across languages for the same claim. As Table 2 shows, **34.5%** of claims have mixed quality, while only **0.6%** are low-quality in all languages. Filtering at 85 would drop many otherwise strong translations.
> > >
> > > **(c) A global threshold removes a large amount of good data.**
> > >
> > > Filtering at 85 would remove **35.0%** of claims, including **220,769 high‑quality translations (28.3%)** that are lost only because another language in the same claim falls below threshold.
> > >
> > > **Table 2: Data Loss at Threshold 85**
> > >
> > > |Category|AnyLang<85|AllLangs<85|MixedQuality|High-QualityDataLost|
> > > |-|-|-|-|-|
> > > |**Count**|27,304|435|26,869|**220,769**|
> > > |**% of Data**|35.0%|0.6%|34.5%|**28.3%**|
> > >
> > > To avoid building a specific quality–coverage trade‑off into the benchmark, we **retain all translations and release per‑claim, per‑language GEMBA scores**, enabling downstream users to define their own strict or standard subsets.
> > >
> > > # Task Feasibility & "Below Chance" Scores
> > >
> > > We fully agree that FEVER is traditionally evidence-dependent. However, Poly-FEVER addresses a distinct research question: **Intrinsic Hallucination Detection.** We are not measuring *verification* (can the model find proof?), but rather *parametric knowledge reliability* (does the model "know" this fact, or does it hallucinate?).
> > >
> > > **Is the task impossible?**
> > > The reviewer expressed concern that removing evidence makes the task impossible (akin to "summarization without text"). To address this empirically, we evaluated recent state-of-the-art models on Poly-FEVER. As shown below, **newer models consistently achieve well above chance performance**, with many exceeding the historical 50.9% English baseline.
> > >
> > > **Performance of Recent Models (Accuracy %)**
> > >
> > > |Model|en|zh-CN|ar|ja|ko|am|
> > > |-|-|-|-|-|-|-|
> > > |**LLaMA-3.2-3B**|50.06|43.77|41.59|40.31|49.42|39.15|
> > > |**Phi-3.5-4B**|**63.17**|46.10|38.23|34.27|48.91|26.70|
> > > |**Mistral-7B**|**65.20**|**52.11**|45.80|40.12|**56.60**|33.41|
> > > |**Gemma3-27B**|**64.18**|**62.9**|49.42|**52.5**|**62.13**|41.21|
> > > |**Qwen-2.5-32B**|**58.54**|**54.94**|**49.42**|**53.15**|46.73|**41.21**|
> > >
> > > The fact that Mistral-7B and Qwen-2.5 achieve respectable accuracy proves the task is **feasible and valid**. It measures the model's internal world model, which is improving over time.
> > >
> > > **Regarding "Below Chance" Scores (e.g., 19%):**
> > > The reviewer correctly noted that LLaMA-2 70B achieved scores like 19.47%, asking if this indicates an evaluation error. We confirm this is **not a technical bug**, but rather a documented failure known as **Label Bias** [1] and **Instruction Drifting** in low-resource settings [2].
> > >
> > > 1.  **Instruction Failure:** In languages like Tamil or Amharic, older models (like LLaMA-2) often fail to adhere to the binary "True/False" constraint, outputting conversational filler or repetitions. These are strictly scored as failures [2].
> > > 2.  **Prior Collapse:** When uncertain, models collapse to a single class (e.g., predicting "True" due to sycophancy [3]). If the specific subset of claims for that language happens to be unbalanced (or if the model collapses to the *minority* class due to token probability artifacts), accuracy drops significantly below 50% [1].
> > >
> > > These low scores accurately reflect that LLaMA-2 70B is **unsafe** for fact-checking in these languages, which is exactly the insight Poly-FEVER aims to provide.
> > >
> > >  [1] Zhao, T., et al. (2021). "Calibrate Before Use: Improving Few-Shot Performance of Language Models." ICML.
> > >
> > > [2] Li, C., et al. (2024). "X-instruction: Aligning language model in low-resource languages with self-curated cross-lingual instructions." ACL Findings.
> > >
> > > [3] Sharma, M., et al. (2024). "Towards Understanding Sycophancy in Language Models." ICLR.

---

### Official Review · Reviewer_gtDy · 2025-10-31

**Soundness:** 3
**Presentation:** 4
**Contribution:** 4
**Rating:** 8
**Confidence:** 4

**Summary:**

The authors introduce PolyFEVER, a multilingual benchmark for fact verification that spans 11 languages. The authors evaluate a set of foundation models and compare their results on the benchmark against language web presence and other variables. The multilingual data is generated by translating English text. The paper is an important step towards AI fairness across high- and low-resource languages.

**Strengths:**

- The argument for developing a multilingual fact verification benchmark is very compelling, and it is highlighted by the results showing that lower-resource languages correlate with worse overall performance on the task. We need more papers in the field tackling this sort of problem, and this paper takes a good stab at enabling this line of work.

- Figure 1 is a nice addition that contextualizes the problem well. This language balance is reasonable and helpful, as is the balance among different topics.

- The presentation of results is comprehensive and informative. The RAG experiments over Wikipedia in particular are interesting.

- The paper is well-written.

**Weaknesses:**

- It would be ideal to include claims originally written in the target languages instead of translating English claims. Translated claims introduces bias, and ignores nuances that would otherwise be modeled if drawing from content that was originally written for audiences who speak these languages.

- The language models used for evaluation are a bit old, especially GPT-3.5. This isn't a critical issue, but the paper may be seen as more timely and receive better reception if some of the experiments are updated to include contemporary models released in 2025.

- Fact verification is intrinsically difficult as there are many natural language claims that are not strictly true or false. This paper partially addresses this by adopting the FEVER approach of including a "not enough info" label, but this fails to capture the nuance of partially subjective or contextual claims. A discussion of this concept could enhance the paper.

**Questions:**

- It would be nice to include more of a discussion w.r.t. future work on methods for this benchmark.

---

> ### Author Response · Authors · 2025-11-21
>
> We sincerely thank the reviewer for the strong rating and for highlighting the "compelling argument" and "comprehensive results" of Poly-FEVER. We address the constructive suggestions below.
>
> # Cross-Lingual Claim Source (W1)
>
> We fully agree that claims natively written in the target languages would add valuable cultural and stylistic diversity. For Poly‑FEVER, however, our goal was to enable **controlled** cross‑lingual comparison. We therefore follow the design of prior multilingual fact‑checking benchmarks, such as X-FACT[1] and XFEVER[2], which translate verified English claims so that each example has a single, shared ground‑truth label across all languages. This makes it possible to attribute performance differences to language and resource effects rather than to differences in which issues are salient or how facts are framed in different communities. We will highlight in the camera‑ready that collecting native‑language claims is an important complementary resource, but that it would constitute a different, less directly comparable task.
>
> [1] Gupta, Ashim, and Vivek Srikumar. "X-fact: A new benchmark dataset for multilingual fact checking." arXiv preprint arXiv:2106.09248 (2021).
>
> [2] Chang, Yi-Chen, Canasai Kruengkrai, and Junichi Yamagishi. "XFEVER: Exploring fact verification across languages." arXiv preprint arXiv:2310.16278 (2023).*
>
> # Update Models (W2)
>
> We agree that the field moves fast. The following table is the extension for 4 updated models on accuracy of hallucination detection, following settings of Section 5.3, L345 - L367. In addition, we refer the reviewer to our response to Reviewer 66Y3, section Retrieval Design for LLaMA 3.2 3B model on retrieval task.
>
> ||en|zh-CN|hi|ar|bn|ja|ko|ta|th|ka|am|
> |-|-|-|-|-|-|-|-|-|-|-|-|
> |LLaMA 3.2 1B|52.76|41.08|43.90|33.50|44.54|29.53|44.93|30.68|34.53|34.53|21.82|
> |LLaMA 3.2 3B|50.06|43.77|43.77|41.59|38.90|40.31|49.42|40.56|40.18|41.85|39.15|
> |Phi 3.5 4B|63.17|46.10|47.62|38.23|48.01|34.27|48.91|35.30|39.79|39.28|26.70|
> |Mistral 7B Instruct|65.20|52.11|55.40|45.80|53.90|40.12|56.60|41.29|45.55|44.83|33.41|
>
> # Factual Nuance (W3)
>
> Poly‑FEVER intentionally discards the “not-enough-information (NEI)’’ label (see Section 3.2) and keeps only Supported/Refuted claims. The aim is to study hallucination tendencies when models must commit to a truth value without access to documents, rather than to model the full spectrum of partial or contextual truth. We agree that real‑world factuality is more nuanced, and will add a short discussion in Section 7 explaining how Poly‑FEVER can be extended with graded or multi‑label judgments.
>
> # Future work enabled by Poly‑FEVER (Q1)
>
> We will expand the conclusion to better articulate follow‑up directions, including:
>
> (i) examining culturally grounded factual reasoning by contrasting translated claims with native-authored ones to study cultural, lexical, and discourse-norm effects, and (ii) developing multilingual safety, robustness, and alignment methods, including cross-language hallucination transfer, consistency-based evaluation, and fact-grounded generation in low-resource settings.
>
>  We appreciate the reviewer’s encouragement along these lines.

---

> > ### Comment · Reviewer_gtDy · 2025-11-23
> >
> > Thank you very much for the thoughtful response! The additional points and data are very much appreciated.

---

### Official Review · Reviewer_66yY · 2025-11-01

**Soundness:** 2
**Presentation:** 3
**Contribution:** 2
**Rating:** 4
**Confidence:** 3

**Summary:**

This paper introduces Poly-FEVER which is a multilingual benchmark extending FEVER, Climate-FEVER, and SciFact to 77,973 labeled claims across 11 languages for evaluating hallucination detection in LLMs. The authors translate English claims using Google Translate, validate translations with GEMBA scores, apply LDA for topic modeling (22 topics), and benchmark several LLMs (ChatGPT-3.5, LLaMA-2, LLaMA-3.1-8B) under various prompt designs. They investigate correlations between accuracy and web presence (Google hit counts) and test retrieval-augmented generation (RAG) using DPR over Wikipedia. Results show performance is difference in high-resource languages, with topic structuring benefiting from lower-resource settings and RAG providing mixed results.

**Strengths:**

1. The paper introduced 77,973 claims across 11 typologically diverse languages is substantial includes high-to low-resource settings and multiple scripts (logographic, alphabetic, abjad). The scale and coverage is fair.
2. Including LDA topic distributions as metadata enables topic-stratified analysis, which is underexplored in prior multilingual NLP benchmarks.

**Weaknesses:**

1. experimental design is a little confused: THe work compared accuracy across languages conflates translation quality, cultural bias in source data (Wikipedia), LLM pretraining distributions, and linguistic properties. Only attributes gaps to resource imbalance but does not control for claim difficulty, domain familiarity, or translation errors.
2.Section 5.4 correlates web hits with accuracy but does not establish causation. Alternative explanations (e.g., translation errors more frequent in low-resource languages) are not ruled out.
3. It Seems Retrieving only English Wikipedia for all languages (Appendix B.2) is a critical flaw. Multilingual Wikipedia or language-specific corpora would be more appropriate.

**Questions:**

Why retrieve English Wikipedia for non-English claims? Did you experiment with multilingual retrieval or language-specific corpora?
Have you controlled for translation quality when correlating web hits with accuracy? Could low-resource languages simply have worse translations that introduce label noise?

---

> ### Author Response · Authors · 2025-11-21
>
> # Experimental Validity (W1)
>
> We thank the reviewer for raising this important concern.
>
> **Claim difficulty.** The benchmark covers a wide difficulty spectrum: claims vary in length (16 to 1.84k characters for English), span 21 diverse topics, and include cases requiring multi-hop reasoning, domain knowledge, or implicit world facts. To ensure the difficulty reflects genuine factual reasoning rather than ambiguity, all claims are binary (true/false) and we remove any unclear or unsupported instances to maintain clean veracity labels.
>
> **Domain familiarity.** We agree that speakers of different languages may be more or less familiar with specific topics (e.g., American football vs. local politics). Rather than a nuisance, this is part of what Poly-FEVER is designed to expose: whether models treat the **same** claim fairly across languages. We will clarify this framing in the paper.
>
> **Translation quality.** We evaluate translation quality using the GEMBA framework[1], where **scores above 85 indicate professional-quality translation** and **scores around 90 correspond to human-level adequacy**[2]. Across the 10 target languages, adequacy scores range from 88.9 to 93.0 (Table 1, L238–248), indicating consistently high fidelity. To give the community maximal flexibility, we will release per-claim translation quality scores for each language so users can filter, reweight, or construct subsets according to their own translation-quality thresholds and experimental needs.
>
> **Web presence vs. accuracy.** In Section 5.4 our goal is descriptive, not causal. The Spearman correlation between average web‑hit counts and accuracy is $\rho ≈ 0.49(p = 0.13)$, which we now explicitly report as a moderate but not statistically significant association. We will soften the text to state that web presence is **one plausible factor** consistent with prior work [3,4] while noting that other factors may also contribute.
>
> [1] Kocmi, Tom, and Christian Federmann. "GEMBA-MQM: Detecting translation quality error spans with GPT-4." arXiv preprint arXiv:2310.13988 (2023).
>
> [2] Freitag, Markus, et al. "Experts, errors, and context: A large-scale study of human evaluation for machine translation." Transactions of the Association for Computational Linguistics 9 (2021): 1460-1474.
>
> [3] Pires, Telmo, Eva Schlinger, and Dan Garrette. "How multilingual is multilingual BERT?." arXiv preprint arXiv:1906.01502 (2019).
>
> [4] Singh, Pranaydeep, Orphée De Clercq, and Els Lefever. "Distilling monolingual models from large multilingual transformers." Electronics 12.4 (2023): 1022.
>
> # Retrieval Design (W2, Q1)
>
> We appreciate the opportunity to clarify why we used English Wikipedia for Dense Passage Retrieval (DPR).
>
> We retrieve Wikipedia passages in each claim's native language, and then intentionally retrieve from **English Wikipedia for all languages** to ensure methodological consistency and a uniform retrieval corpus. By fairness, we refer specifically to **equalizing the retrieval domain** so that models across languages are grounded in the same evidence source, avoiding advantages caused by richer or more frequently updated corpora in high-resource languages.
>
> Results for native vs. English Wikipedia retrieval (LLaMA-3.2-3B) with source size and document passages are shown below:
>
> ||en|zh-CN|hi|ar|bn|ja|ko|ta|th|ka|am|
> |-|-|-|-|-|-|-|-|-|-|-|-|
> |Src_size|74G|5.3G|985M|5.5G|1.4G|12G|2.2G|1.3G|1.9G|835M|31M|
> |Doc_passages|21,015,300|5,855,162|948,985|6,421,656|1,201,155|5,256,853|2,986,027|791,163|950,254|599,116|31,489|
> |WithoutRAG|50.06|43.77|43.77|41.59|38.90|40.31|49.42|40.56|40.18|41.85|39.15|
> |RAG_native|51.09|46.72|45.70|44.93|44.29|40.82|45.06|39.67|44.80|44.03|39.28|
> |RAG_en|51.08|48.69|42.90|52.89|46.92|41.75|50.00|39.35|47.37|46.34|39.12|
>
> This experiment confirms that mixed-language retrieval introduces comparability issues. Dense-retrieval models, tokenizer tools, and high-quality cleaned Wikipedia dumps are **not uniformly available** across languages.
>
> Using language-specific corpora would therefore introduce uncontrolled variation in **coverage**, **article quality**, and **update frequency**. Fixing the retrieval corpus to English ensures a **uniform factual grounding source**, so that cross-lingual performance differences primarily reflect **model reasoning and linguistic generalization**, rather than retrieval-domain imbalance[5,6].
>
> This also highlights a core motivation of Poly-FEVER: even with retrieval-based mitigation, most languages must rely on **English-centric retrieval infrastructure**, because robust multilingual retrieval resources remain limited or unavailable. We will clarify this rationale in the revision.
>
> [5] Zhang, Xinyu, et al. "Mr. TyDi: A multi-lingual benchmark for dense retrieval." arXiv preprint arXiv:2108.08787 (2021).
>
> [6] Zhang, Xinyu, et al. "Miracl: A multilingual retrieval dataset covering 18 diverse languages." Transactions of the Association for Computational Linguistics 11 (2023): 1114-1131.

---

### Meta-Review · Area_Chair_B9mc · 2026-01-06

**Summary:**

This submission introduces Poly-FEVER, a multilingual benchmark derived from FEVER/Climate-FEVER/SciFact, containing ~77.9K parallel claims across 11 languages with topic metadata and a suite of baseline evaluations (prompting variants, topic-augmentation, web-presence correlation, and DPR-based retrieval augmentation).

Reviewers generally agree the motivation and dataset scale are valuable, with one reviewer strongly supportive (accept) and others raising critical concerns about evaluation validity and claim overreach, leading to divergent scores. The major concerns that lead to AC's decision are: 1) the less updated results on more advanced models, such as proprietary models, and more hallucination detection methods; 2) robustness to translation; and 3) missing sanity checks to rule out evaluation artifacts behind extreme below-chance accuracies.

**Reviewer Concerns:**

The rebuttal largely addresses concerns about clarity and positioning: it clarifies the Table 3/Section 5.5 mismatch and Figure 2 intent (Thxp/W6/W9/Q2), agrees to soften or move low-value analyses like the temperature sweep to the appendix (W7), provides a rationale and some evidence for the English-vs-native retrieval design (66yY Q1/W2), adds newer-model results (gtDy W2), and reframes the web-hit analysis as descriptive and (by their report) not statistically significant (66yY/W3). Still outstanding are the core validity and integrity issues raised by 8K2G (and partially 66yY): the paper needs concrete sanity checks to rule out evaluation artifacts behind extreme below-chance accuracies (invalid-output rates, label balance baselines, confusion/per-class stats), tighter task framing so it’s clearly “sentence-only parametric belief probing” rather than evidence-based verification (especially for SciFact), alignment of abstract/contribution claims with the non-significant web-presence result, and stronger translation-quality reporting beyond mean GEMBA (tail/distribution and inter-annotator agreement for the manual audit).

**Reviewer Scores:**

I think the scores might remain the same for all reviewers.

---

### Decision · Program_Chairs · 2026-01-26

Reject